# Nutritional Partitioning among Sympatric Ungulates in Eastern Tibet

**DOI:** 10.3390/ani14152205

**Published:** 2024-07-30

**Authors:** Zhengwei Luo, Chao Pei, Haonan Zhang, Yichen Wang, Baofeng Zhang, Defu Hu

**Affiliations:** School of Ecology and Nature Conservation, Beijing Forestry University, Beijing 100083, China; luozw114688@gmail.com (Z.L.); chaopei@bjfu.edu.cn (C.P.); zhanghn911@163.com (H.Z.); wangyichen125@163.com (Y.W.); baofengdaisy@126.com (B.Z.)

**Keywords:** dietary analysis, DNA barcode, sympatric ungulate, niche overlap, alpine musk deer, white-lipped deer, red serow

## Abstract

**Simple Summary:**

Alpine musk deer, red serow, and white-lipped deer coexist in the Nyenchen Tanglha Mountains of Tibet. We aimed to understand the mechanisms of their coexistence by studying their dietary preferences using DNA barcoding. All of the species exhibited broad dietary ranges with distinct food preferences. Furthermore, our findings revealed genus-level dietary specializations and the mechanisms facilitating their coexistence. The results of this study provide valuable insights for the development and implementation of effective conservation strategies and management measures in the local area.

**Abstract:**

Wild ungulates play crucial roles in maintaining the structure and function of local ecosystems. The alpine musk deer (*Moschus chrysogaste*), white-lipped deer (*Przewalskium albirostris*), and red serow (*Capricornis rubidus*) are widely distributed throughout the Nyenchen Tanglha Mountains of Tibet. However, research on the mechanisms underlying their coexistence in the same habitat remains lacking. This study aimed to investigate the mechanisms underlying the coexistence of these species based on their dietary preferences through DNA barcoding using the fecal samples of these animals collected from the study area. These species consume a wide variety of food types. Alpine musk deer, white-lipped deer, and red serow consume plants belonging to 74 families and 114 genera, 62 families and 122 genera, and 63 families and 113 genera, respectively. Furthermore, significant differences were observed in the nutritional ecological niche among these species, primarily manifested in the differentiation of food types and selection of food at the genus level. Owing to differences in social behavior, body size, and habitat selection, these three species further expand their differentiation in resource selection, thereby making more efficient use of environmental resources. Our findings indicate these factors are the primary reasons for the stable coexistence of these species.

## 1. Introduction

Wild ungulates often play an important role in maintaining the structure and function of ecosystems and are commonly regarded as one of the ecological indicators of the health status of forest ecosystems [1]. Furthermore, wild ungulates are often important components of grassland and woodland food webs and frequently exert a certain degree of direct and indirect influence on the composition, structure, energy flow, and community succession patterns of forest vegetation through feeding, trampling, and excretion [2]. In addition, a widespread decrease in the population of top predators and the implementation of increasingly stringent animal conservation measures may contribute to an increase in the number of wild ungulates in the same area, thereby potentially leading to intensified intra- and inter-species competition [3]. Moreover, wild ungulates can potentially damage farmland and compete with livestock for resources, which may in some cases escalate human–wildlife conflicts. Notably, intense intra- and inter-species competition, along with potential human–wildlife conflicts, may pose new challenges for conservation and management efforts [4].

The mechanism underlying the coexistence of closely related species in ecological niches is an important research topic in animal ecology. Studying the occurrence and maintenance mechanisms of species coexistence is important for community ecology research [5]. Various hypotheses and theories have been proposed to elucidate the mechanisms of species coexistence, including the niche differentiation hypothesis [6], neutral theory [7], adaptive boundary theory [8], environmental heterogeneity hypothesis [9], and the competitive exclusion principle [10]. The competitive exclusion principle has been widely applied in the study of coexistence mechanisms [11]. Competition occurs in environments with limited resources if two or more species share the same resources and have a significant ecological niche overlap. The core concept of competitive exclusion is competitive pressure, and in the presence of intense competition between species, species that are better adapted to resource utilization have a higher chance of survival and reproductive success. Meanwhile, species that are less adapted to resource utilization may be excluded from habitats or forced to seek alternative resource-utilization strategies or habitats [12]. Furthermore, competitive exclusion is closely related to niche differentiation. Niche differentiation occurs in environments with intense competition, wherein species avoid direct competition by adapting to different resource-utilization strategies or occupying different ecological niches, thereby achieving coexistence [13]. This differentiation can be achieved through adaptive changes such as species morphology, behavior, and feeding habits. For example, many studies have shown significant overlap in food resources among coexisting species [14]. Investigating the dietary composition and nutritional niche overlap among small, medium, and large ungulate species within the same habitat can reveal the occurrence and maintenance mechanisms of species coexistence within food webs [15]. This suggests that food, as a crucial resource, can be shared and competed for, and it fluctuates in availability. Understanding the adaptive strategies that allow these ungulate species to coexist despite overlapping resource use is an important part of understanding their community dynamics.

Alpine musk deer (*Moschus chrysogaste*) [16] and red serow (*Capricornis rubidus*) [17] are classified as browsers, whereas white-lipped deer (*Przewalskium albirostris*) [18] do not exhibit a clear browsing preference. All three species primarily feed on plants, and their nutritional niches are relatively similar. In the wild, these species often exhibit overlapping ranges. Exploring the relationship between feeding habits and nutritional niches is important for the conservation and management of forest ungulates, as well as for revealing the mechanisms of coexistence among ungulate species in the same habitat.

Many methods are used to study wildlife diets. The simplest traditional approach is direct observation of feeding behavior, which provides valuable information about an animal’s diet. However, observing the feeding habits of some elusive, timid, and nocturnal wildlife species is challenging in the wild, requiring substantial effort and resources [19]. Consequently, alternative methods have emerged, such as food residue analysis, stomach content examination, fecal microanalysis, and stable isotope analysis. The fecal microanalysis technique, which offers advantages like convenient sampling, non-invasiveness, and lower cost, has been widely used to study wildlife dietary habits since the 1970s [20]. Yet, this method has limitations—it requires experienced examiners for microscopic observation and is time- and labor-intensive. Additionally, the identification resolution may be low due to morphological similarities in plant cells and variations in animal digestive processes toward different plant species [21].

With the advancement of molecular biology, researchers have employed molecular biology techniques to analyze wildlife fecal samples, overcoming the limitations of traditional dietary research methods. However, these traditional methods often suffer from low resolution and high cost and effort [21]. The new molecular biology approach allows for accurate, rapid, and straightforward acquisition of animal dietary data, to a certain extent compensating for the shortcomings of traditional methods. DNA barcoding [22,23,24,25,26] has already been applied in some studies on the dietary composition of wild animals.

Three ungulate species, alpine musk deer, white-lipped deer, and red serow, coexist in the Tibet Nyenchen Tanglha Mountains; however, research on the mechanisms underlying their coexistence is lacking. We investigated the dietary composition of these three ungulate species, determined whether there could be significant dietary partitioning or overlap among them, and contributed to the understanding of whether dietary composition is one of the mechanisms underlying their coexistence in this area. We hypothesize that the three species of ungulates exhibit distinct nutritional partitioning in their diets. This nutritional partitioning serves as one of the mechanisms facilitating their coexistence

## 2. Materials and Methods

### 2.1. Study Area

The study area (93.906983° E–94.812076° E and 31.462707° N–31.036659° N) is located in Eastern Tibet, northwest of Chamdo City, at the northern foot of the Nyenchen Tanglha Mountains. This region is characterized by rugged mountains with an average elevation of over 4000 m and is adjacent to the Nujiang River system. The climate type of the area is highland temperate semi-humid. The warm season in the study area is from July to October, with average temperatures above 8–15 °C. This is also the period when thunderstorms and hailstorms are more common in the region. The vertical zonation is significant in this area. Owing to limited human activities and spontaneous animal conservation efforts by the Tibetan people because of religious reasons, this region has abundant flora and fauna. In addition to the alpine musk deer, red serow, and white-lipped deer, this region also harbors other animal species, such as Tibetan brown bears (*Ursus arctos pruinosus*), snow leopards (*Panthera uncia*), and wild yaks (*Bos mutus*). The forest types in the research area are mainly mixed coniferous and broad-leaved forests, as well as pure coniferous forests. The main tree species include Asian white birch (*Betula platyphylla*), purple cone spruce (*Picea purpurea*), Chinese weeping cypress (*Cupressus funebris*), willow (*Salix sp.*), and Sikang pine (*Pinus densata*).

### 2.2. Sample Collection and Preservation

To investigate whether the three ungulate species display significant dietary differences to alleviate competition pressures, we collected fecal samples from alpine musk deer, white-lipped deer, and red serow in the study area. Due to the high elevation of the research area, during cold climate periods, fecal samples can be easily covered by heavy snow, which would create significant difficulties and dangers for sample collection. Therefore, we have chosen to conduct the fecal sample collection during the period from 20 July 2023 to 3 October 2023. Twelve vertical transect lines, with a total length of 72.35 km, were designed through communication with local Tibetan guides. These transects spanned an elevation range of 3600 to 4200 m, encompassing the major habitat types present within the study area. The sampling area exhibited distinct vertical zonation, with habitat types arranged from lower to higher elevations as follows: mixed coniferous and broad-leaved forests, coniferous forests, alpine shrub meadows, and alpine meadows. The habitat type of each collected sample was determined by combining this information with photographs of the sampling sites. The collected fecal samples were labeled with the following information: date of collection, freshness, coordinates, and elevation. The fecal samples were placed in 25 mL sterile centrifuge tubes with color-changing silica gel and stored in a −20 °C freezer for preservation. In total, 170 suspected ungulate fecal samples were collected. In order to provide some reference materials for the subsequent identification of dietary plants, we also collected plant samples from the vicinity of the fecal sampling sites. The plant collection bag consisted of size 10 self-sealing bags, Kraft paper envelopes, and color-changing silica gel. The plant samples were stored in well-ventilated, cool, dry places.

### 2.3. Species Identification from Fecal Samples

To conduct species identification on the collected fecal samples, we carried out the following procedures: Three milliliters of the fecal sample was transferred into a 15 mL centrifuge tube with 5 mL of PBS (phosphate buffered saline), and the tubes were incubated for 2 min with vigorous shaking to facilitate the detachment of animal intestinal cells from the fecal surface. Total DNA was extracted from the fecal samples using a TIANamp Genomic DNA Kit (Cat. No. 4992254; Tiangen, Beijing, China). DNA extraction results were examined using 1% agarose gel electrophoresis. The extracted DNA was stored in a freezer at −20 °C. The universal primers 16S-F/R [27], designed for vertebrates, were used to amplify an approximately 550 bp fragment of the ungulate mitochondrial gene [27]. Each reaction system of total reaction volume of 25 μL contained 3 μL DNA template, 12.5 μL 2× Premix Taq (Tiangen), 1 μL each forward and reverse primer (10 μM), 3.5 μL ddH_2_O, and 4 μL bovine serum albumin (20 μg/μL, A8010, Solarbio, Beijing, China). The polymerase chain reaction (PCR) conditions included an initial denaturation at 95 °C for 5 min, followed by 30 cycles of denaturation at 94 °C for 30 s, annealing at 49 °C for 30 s, and extension at 72 °C for 45 s, followed by a final extension at 72 °C for 5 min. Successfully amplified DNA samples were sequenced (SinoGenoMax Limited Company, Beijing, China) using an ABI 3730XL sequencing instrument (Applied Biosystems Inc., Foster City, CA, USA). The raw sequences were trimmed and assembled to obtain aligned sequences approximately 126 bp in length using Geneious Prime 2022.0.1 (Biomatters Ltd., Auckland, New Zealand). Using the Basic Local Alignment Search Tool (BLAST) provided by the National Center for Biotechnology Information (NCBI), the alignment-ready sequences were aligned with the GenBank database for online species matching. The fecal sample was identified as originating from the species corresponding to the best-matching sequence, with a coverage of 100% and a similarity of ≥98% with the query sequence.

### 2.4. Dietary Identification

#### 2.4.1. DNA Extraction

In order to ensure the dietary identification results accurately reflected the animals’ diets during the study period, we selected fresh fecal samples (with moist fecal surfaces) from the successfully identified samples to perform DNA extraction. An E.Z.N.A. Soil DNA Kit (Omega Bio-tek, Inc., Norcross, GA, USA) was used to extract genomic DNA from the fecal samples. The quality and concentration of the DNA were measured using a Nanodrop 2000 spectrophotometer (Thermo Fisher Scientific, Inc., Waltham, MA, USA). The DNA samples were stored at −20 °C.

#### 2.4.2. PCR

The chloroplast rbcl2 region was amplified using the universal primers rbcl2 F/R (5′-CTTACCAGYCTTGATCGTTACAAAGG-3′)/(5′-GTAAAATCAAGTCCACCRCG-3′). To distinguish between different samples, an 8 bp barcode sequence was added to the 5′ end of both the upstream and downstream primers. The synthesized universal primers with barcode sequences were amplified using an ABI 9700 PCR instrument (Applied Biosystems Inc., Foster City, CA, USA). The PCR amplification system consisted of 2 μL DNA template (total DNA 30 ng), 1 μL each of forward and reverse primers (5 μM each), 3 μL bovine serum albumin (2 ng/μL), 12.5 μL 2× Taq Plus Master Mix, and 5.5 μL ddH_2_O, resulting in a total reaction volume of 25 μL. The PCR conditions were as follows: initial denaturation at 94 °C for 5 min, followed by 35 cycles of denaturation at 94 °C for 30 s, annealing at 55 °C for 30 s, and extension at 72 °C for 60 s, followed by a final extension at 72 °C for 7 min. The amplified PCR products were analyzed for band size using 1% agarose gel electrophoresis at 170 V for 30 min. The purified PCR products were subjected to automated purification using Agencourt AMPure XP (Beckman Coulter, Inc., Indianapolis, IN, USA). Sequencing was performed on an Illumina Miseq/NovaSeq 6000 platform (Illumina, Inc., Hayward, CA, USA) using a paired-end (PE) sequencing strategy with read lengths of 250 (PE250) or 300 bases (PE300).

### 2.5. Dietary Data Analysis

The sequencing results yielded PE sequencing data. The FASTQ data obtained were subjected to quality control processing to obtain high-quality data. The FASTQ data were split into different samples based on the barcode sequences. Pear software (version 1.8.0) was used for quality control of the FASTQ data by removing sequences with ambiguous bases and primer mismatches. The sequences were trimmed to remove bases with quality values <Q20. The PE reads were merged based on their overlapping relationship, with a minimum overlap of 10 bp and a *p*-value cutoff of 0.0001. This process generated FASTA sequences, which were used to remove chimeric and short sequences using Vsearch software (version 2.15.0). Subsequently, the high-quality sequences were subjected to operational taxonomic unit (OTU) clustering with a sequence similarity threshold of 97%. To assign taxonomic information to each OTU, a sequence alignment against the NCBI database was performed using the BLAST algorithm.

The principles for species identification based on sequence comparison were as follows: (1) When the identity in the comparison results was <95%, it was recorded as unidentified. (2) When the identity in the comparison results was >95% and no species were recorded in the matched local species distribution, the species with the highest identity was selected. The identification result was recorded as the lowest taxonomic unit that encompassed all species with the highest identity and was consistent with local species records. (3) When the identity in the comparison results was >95% and <98%, the species with the highest identity, which also matched the local species distribution records, was selected. The identification results were recorded as the lowest taxonomic unit that encompassed all local species with the highest identity. (4) When the identity in the comparison results was >98% and multiple species matched the local distribution records, identification was made at the genus level; if only one species matched the local distribution records, identification was made at the species level; if no species matched the local distribution records, identification was made at the genus level, indicating that the identified species belonged to the same genus as those recorded in the local distribution. Based on these principles, each sequence was subjected to species identification. Species identification information was obtained from iplant (https://www.iplant.cn/) (accessed on 28 October 2023), iflora (http://www.iflora.cn/) (accessed on 28 October 2023), and species 2000 (http://col.especies.cn/) (accessed on 28 October 2023), as well as from a plant catalog compiled based on plant collection and identification within the research area.

### 2.6. Data Statistical Analysis

The spatial overlap among the three species was quantified using the Minimum Convex Polygon (MCP) method, which helped delineate their spatial distributions and assess the potential for coexistence. The Shannon, Simpson, and Pielou’s evenness indices were calculated to measure the biodiversity of the three species. Niche breadth was calculated using Levins’ niche breadth index. Relative abundance (abundance of a species in a sampling unit/total abundance of all species in a sampling unit) was used to measure the dietary habits of the three species. Ecological niche overlap was measured using Pianka’s overlap index. Individual specialization was assessed using the individual specialization index (the ratio of average individual niche breadth to total population niche breadth). The individual specialization index is a dimensionless index that ranges from one, when all individuals consume the same prey in the same proportions (no individual specialization), down to zero, when each individual uses a unique type of prey (maximal individual specialization) [28]. AMOVA (Analysis of Molecular Variance) was used to detect significant differences in dietary patterns among the species. ANOVA (Analysis of Variance) was employed to examine significant variations in biodiversity indices, niche breadths, and elevation ranges. Kruskal–Wallis analysis was utilized to identify significant differences in food types consumed. ANOSIM (Analysis of Similarities) was applied to determine significant differences in dietary compositions between and within species.

## 3. Results

### 3.1. Spatial Distribution

A total of 35 fecal samples from alpine musk deer were successfully identified. Of these, 6 were collected from alpine meadows, 20 from alpine shrub meadows, 7 from coniferous forests, and 2 from mixed coniferous and broad-leaved forests. Additionally, 22 samples were collected from white-lipped deer, including 2 from alpine meadows, 14 from alpine shrub meadows, 5 from coniferous forests, and 1 from mixed coniferous and broad-leaved forests. For red serow, 15 samples were collected, with 7 from alpine shrub meadows, 4 from coniferous forests, and 4 from mixed coniferous and broad-leaved forests (Figure 1a).

To investigate the spatial distribution relationships among the alpine musk deer, red serow, and white-lipped deer, a 3D scatter plot (Figure 1b) was generated based on the latitude, longitude, and elevation data of the sampling locations. The spatial overlap among the three species was quantified using the minimum convex polygon (MCP) method. The spatial overlap index between the alpine musk deer and red serow was 0.29, between the red serow and white-lipped deer was 0.53, and between the alpine musk deer and white-lipped deer was 0.48. The spatial overlap data and the 3D scatter plot visually demonstrate substantial spatial overlap among the three species. The high overlap values between the red serow and white-lipped deer (0.53), as well as between the alpine musk deer and white-lipped deer (0.48), indicate that these species occupy the same geographical region and exhibit significant spatial overlap. This suggests the potential for intense interspecific competition and ecological interactions within the community.

### 3.2. Food Composition

We utilized DNA barcoding to reveal the dietary habits of the three ungulate species within the study area over a 2.5-month period during the summer of 2023. DNA barcoding revealed that the white-lipped deer consumed a diet consisting of plants belonging to 62 families and 122 genera. Red serow consumed a diet consisting of plants belonging to 63 families and 113 genera. Alpine musk deer consumed a diet consisting of plants belonging to 74 families and 144 genera. The overlapping portions of the diets included plants belonging to 44 families and 62 genera. Relative abundance stacked bar plots and standard error of dietary intake among individual plots were generated based on dietary data at the family and genus levels in the fecal samples (Figure 2a–d).

At the family and genus levels, the sum of plants (families/genera) with relative abundances exceeding 1% accounted for 94.6/92.0%, 87.4/82.7%, and 89.3/86.3% of the dietary plants of the alpine musk deer, white-lipped deer, and red serow, respectively. The alpine musk deer diet encompassed plants belonging to 9 families and 12 genera. At the family level, the composition of the diet for the alpine musk deer included *Rosaceae* (46.9%), *Onagraceae* (15.6%), *Polygonaceae* (12.8%), *Grossulariaceae* (7.5%), *Crassulaceae* (3.8%), *Asteraceae* (2.3%), *Ranunculaceae* (1.6%), *Caprifoliaceae* (1.4%), *Betulaceae* (1.3%), and *Ericaceae* (1.3%). At the genus level, the diet composition of the alpine musk deer included *Rosa* (27.3%), *Chamerion* (15.6%), *Bistorta* (12.4%), *Fragaria* (9.8%), *Cotoneaster* (8.5%), *Ribes* (7.5%), *Rhodiola* (3.3%), *Artemisia* (2.3%), *Anemone* (1.6%), *Lonicera* (1.3%), *Rhododendron* (1.3%), and *Betula* (1.3%). The white-lipped deer diet encompassed plants belonging to 15 families and 20 genera. At the family level, the composition of the diet for the white-lipped deer included *Salicaceae* (22.8%), *Polygonaceae* (12.6%), *Poaceae* (8.4%), *Cyperaceae* (8.0%), *Rosaceae* (5.7%), *Caprifoliaceae* (5.0%), *Asteraceae* (5.0%), *Grossulariaceae* (4.8%), *Betulaceae* (3.7%), *Onagraceae* (3%), *Mniaceae* (2.5%), *Lamiaceae* (2.3%), *Ulmaceae* (1.4%), *Cephaloziaceae* (1.1%), and *Amaranthaceae* (1.1%). At the genus level, the composition of the diet for the white-lipped deer included *Salix* (22.8%), *Carex* (8.0%), *Koenigia* (7.7%), *Bistorta* (4.9%), *Ribes* (4.8%), *Lonicera* (4.2%), *Artemisia* (3.9%), *Betula* (3.7%), *Elymus* (3.5%), *Chamerion* (3.0%), *Calamagrostis* (2.5%), *Pohlia* (2.4%), *Rosa* (2.4%), *Cotoneaster* (1.7%), *Ulmus* (1.4%), *Piptatherum* (1.4%), *Salvia* (1.1%), *Phlomoides* (1.1%), *Cephaloziella* (1.1%), and *Chenopodium* (1.1%). The red serow diet encompassed plants belonging to 12 families and 14 genera. At the family level, the composition of the diet for the red serow included *Rosaceae* (42.7%), *Grossulariaceae* (10.7%), *Onagraceae* (6.1%), *Betulaceae* (5.6%), *Asteraceae* (5.1%), *Santalaceae* (5.0%), *Caprifoliaceae* (4.5%), *Cyperaceae* (2.9%), *Polygonaceae* (2.3%), *Brachytheciaceae* (1.8%), *Hylocomiaceae* (1.4%), and *Mniaceae* (1.0%). At the genus level, the composition of the diet for the red serow included *Sorbus* (21.9%), *Rosa* (14.7%), *Ribes* (10.7%), *Epilobium* (5.9%), *Betula* (5.6%), *Arceuthobium* (5.0%), *Artemisia* (4.8%), *Lonicera* (4.5%), *Spiraea* (3.3%), *Carex* (2.9%), *Prunus* (2.4%), *Bistorta* (1.8%), *Brachythecium* (1.8%), and *Pohlia* (1.0%).

Among the plant families and genera with relative abundances exceeding 1%, seven families and six genera were consumed by all three ungulates. These accounted for combined proportions of 87.9/52.1%, 39.9/23.9%, and 77.0/42.0% (family/genus) of the dietary plants of the alpine musk deer, white-lipped deer, and red serow, respectively. According to the AMOVA (Analysis of Molecular Variance) analysis, the plants commonly consumed by the three ungulate species exhibited significant differences at the genus level (Among df = 2, *p* < 0.01). The shared plant families among the alpine musk deer vs. white-lipped deer vs. red serow included *Rosaceae* (46.9% vs. 5.7% vs. 42.7%), *Onagraceae* (15.6% vs. 3% vs. 6.1%), *Polygonaceae* (12.8% vs. 12.6% vs. 2.3%), *Grossulariaceae* (7.5% vs. 4.8% vs. 10.7%), *Asteraceae* (2.3% vs. 5% vs. 5.1%), *Caprifoliaceae* (1.4% vs. 5% vs. 4.5%), and *Betulaceae* (1.3% vs. 3.7% vs. 5.6%), respectively. The shared plant genera among the alpine musk deer, white-lipped deer, and red serow included *Rosa* (27.3% vs. 2.4% vs. 14.7%), *Bistorta* (12.4% vs. 486% vs. 1.8%), *Ribes* (7.5% vs. 48.0% vs. 10.7%), *Artemisia* (2.3% vs. 3.94% vs. 4.8%), *Lonicera* (1.3% vs. 41.9% vs. 4.5%), and *Betula* (1.3% vs. 3.75% vs. 5.6%), respectively.

### 3.3. Food Types and Ecological Niche

The diets of alpine musk deer, white-lipped deer, and red serow mainly consisted of trees, shrubs, herbs, ferns, and mosses, with other food types accounting for <1% (Figure 3a). Significant differences were observed in the food types of tree and moss among the three species based on the Kruskal–Wallis analysis (among df = 2, *p* < 0.05). The diet of the alpine musk deer mainly consisted of herbs (50.6%) and shrubs (47.5%). The diet of the white-lipped deer mainly consisted of herbs (48.5%), followed by shrubs (37.2%) and a moderate amount of mosses (7.5%) and trees (6.5%). The diet of red serow mainly consisted of shrubs (35.2%) and trees (32.4%), with herbs (26.1%) as a secondary food source. Analysis of the elevation of the sampling points using ANOVA revealed that the elevation of sample point locations from the white-lipped deer and alpine musk deer was similar and significantly higher than that of the red serow (among df = 1, *p* < 0.05) (Figure 3b). Based on the analysis of food diversity and ecological niche width of the three animal species (Table 1 and Figure 3c,d), the Shannon index, the Simpson index, and Pielou’s evenness of alpine musk deer were lower than those of white-lipped deer and red serow, whereas the ecological niche width of the alpine musk deer was smaller than that of the white-lipped deer and red serow. Individual specialization indices were calculated based on individual and population ecological niche width data. Alpine musk deer had the highest individual specialization index (0.53), followed by red serow (0.41) and white-lipped deer (0.36).

Based on genus-level dietary data, the nutritional niche overlap indices among alpine musk deer, white-lipped deer, and red serow were calculated using the Pianka index. The highest overlap index was observed between alpine musk deer and red serow (0.384), followed by red serow and white-lipped deer (0.248) and alpine musk deer and white-lipped deer (0.166).

Non-metric multidimensional scaling (NMDS) analysis demonstrated compositional differences in the genus-level dietary composition among alpine musk deer, white-lipped deer, and red serow (Figure 4a). The horizontal and vertical ranges of fecal samples for the alpine musk deer were −2.08–0.13 and −0.77–1.22, respectively. For the white-lipped deer, the horizontal and vertical ranges of fecal samples were −0.36 to 1.09 and −0.66 to 1.16, respectively. Lastly, for the red serow, the horizontal and vertical ranges of fecal samples were −1.50 to 0.54 and −1.40 to −0.35, respectively. These high non-metric fit (r^2^ = 0.958) and linear fit (r^2^ = 0.817) values indicate that the NMDS analysis has very good quality (Figure 4b). NMDS and ANOSIM analyses revealed a significant difference in the genus-level dietary composition among the three animal species (*p* < 0.01).

## 4. Discussion

To study the diet and nutritional niche of alpine musk deer, white-lipped deer, and red serow, we employed DNA barcoding technology to determine the dietary compositions of the three animal species. This study aimed to explore the coexistence of alpine musk deer, white-lipped deer, and red serow from the perspective of food resource utilization. The experimental results showed that within the study area, white-lipped deer, red serow, and alpine musk deer consumed plants belonging to 62 families and 122 genera, 63 families and 113 genera, and 74 families and 144 genera, respectively, indicating highly diverse diets of these three ungulate species. Compared with previous studies, a significant increase was observed in the number of plant species consumed at the family and genus levels by these three animal species in this study [16,29,30], which may be attributed to improvements in DNA barcoding technology in terms of the accuracy of identification of dietary species [21]. Furthermore, sampling was conducted between July and October, which corresponds to the warm season in the study area. During this period, a rich variety of plant species exists owing to the favorable water and thermal conditions, and animals have more opportunities to choose different foods and expand their dietary range by consuming different types of food owing to the abundant food resources in the environment [31].

In an ecosystem, different species or individuals often select different food resources to reduce direct competition and maximize the utilization of available resources. This differentiated resource utilization allows species or individuals to establish their own niches and resource-utilization strategies within an environment, thereby promoting species diversity and ecosystem stability. To investigate the extent of nutritional niche differentiation among alpine musk deer, red serow, and white-lipped deer, the analysis was conducted in terms of four aspects. First, the dietary food type results revealed the significant differences in the consumption of trees and moss among the three ungulates, indicating a certain level of dietary partitioning in their utilization of food resources (Kruskal–Wallis analysis, among df = 2, *p* < 0.05). Second, the distances and clustering patterns among the samples were visually evident based on the visualization results of the NMDS plot. The three ungulates exhibited significant differences in their dietary composition at the genus level (AMOVA, among df = 2, *p* < 0.01). Furthermore, the dietary differences among species were significantly greater than the differences within species (ANOSIM, *p* < 0.01). Third, the sum of the relative abundances of the top three genera in the diets of the three ungulate species accounted for >38.5% of the total dietary composition. Among the top three genera in terms of relative dietary abundance, only *Rosa* was consumed by both the alpine musk deer and red serow. At the genus level, significant differences were observed in the preferences for these plants among the three animal species. Fourth, based on the results of the nutritional niche overlap index, the highest and lowest values were 0.384 and 0.166, respectively, indicating a relatively low degree of overlap [32] and no significant nutritional niche overlap among alpine musk deer, red serow, and white-lipped deer. Summarizing the conclusions from these four aspects, we believe that during the warm season in this area, the differentiation of the nutritional niches among alpine musk deer, red serow, and white-lipped deer is promoted by the selection of different food types and plant genera by these species, which ultimately reduces the overlap of nutritional niches and helps avoid conflicts resulting from feeding competition.

Based on the data of dietary genera with relative abundances > 1%, comparing the sum of relative abundances, alpine musk deer (92.0%) have a higher sum than white-lipped deer (82.7%) and red serow (86.3%). However, when comparing the number of dietary genera, the number for alpine musk deer (12 genera) was lower than that of white-lipped deer (20 genera) or red serow (14 genera). This indicates that alpine musk deer exhibit a higher selectivity toward certain dietary genera (such as *Rosa*, *Chamerion*, and *Bistorta*). In contrast, white-lipped deer and red serow tended to utilize a more diverse and even range of dietary genera. This is consistent with the results of Pielou’s evenness index and also explains the findings of the diversity analysis, where alpine musk deer consumed a higher number of dietary genera than that of red serow and white-lipped deer but exhibited lower Shannon’s index, Simpson’s index, and niche breadth values compared with those for red serow and white-lipped deer. Further discussing the food selection strategy of the alpine musk deer, we propose the following possible explanations: (1) Alpine musk deer are browser species [16] and tend to prefer tender and nutrient-rich parts of plants that are easily digestible to meet their nutritional requirements. (2) The alpine musk deer is a solitary animal, facing less intraspecific competition than social animals, thereby exhibiting more opportunities to select preferred food. In contrast, red serow and white-lipped deer are social animals that share limited resources, leading to greater resource competition and pressure. Consequently, individuals in the population need to adapt and utilize a wider ecological niche and exhibit a higher degree of specialization to access a greater variety of food resources to meet the nutritional requirements of the entire group, which is consistent with the results of the ecological niche width and individual specialization indices [33]. (3) White-lipped deer and red serow have significantly larger body sizes than alpine musk deer. A larger body size means they need to consume a greater quantity of food. The amount provided by just a few dietary items would be insufficient to meet the feeding requirements of the larger-bodied animals. Therefore, white-lipped deer and red serow tend to select a more diverse array of food sources as their primary dietary components in order to fulfill their nutritional needs [34].

Furthermore, the differences in food types among the three animals, to some extent, may reflect ecological niche differentiation between species. The study area exhibits a clear vertical vegetation distribution. Below an altitude of 3900 m, coniferous forests and mixed coniferous and broad-leaved forests are the dominant vegetation types. From 3900 m to 4200 m, alpine shrub meadows and alpine meadows are the primary vegetation types. Alpine musk deer [35] primarily inhabit coniferous forests and alpine shrubs. White-lipped deer are mainly active in high mountain grasslands during the summer, whereas red serow typically inhabit forested areas. Analysis of food types indicated that red serow primarily consumed trees, shrubs, and herbs, whereas white-lipped deer and alpine musk deer mainly fed on shrubs and herbs. By comparing the altitudes of the fecal sample collection points for the three species, we found that the collection points for alpine musk deer and white-lipped deer were close in altitude and higher than those for red serow. Therefore, we speculate that the red serow has a lower altitude distribution than alpine musk deer and white-lipped deer, indicating a certain degree of spatial ecological niche differentiation among the three species. The study conducted by Shi [36], which placed infrared cameras in the Yarlung Zangbo Grand Canyon, also showed that alpine musk deer were distributed at higher altitudes than red serow. Nevertheless, the spatial distribution relationship between alpine musk deer and white-lipped deer requires further investigation.

## 5. Conclusions

This study concluded that, during the warm season, alpine musk deer, white-lipped deer, and red serow in the study area have a wide variety of food choices. Furthermore, our findings revealed a clear differentiation in the nutritional ecological niche among the three species, which was primarily manifested in the differentiation of food type selection and selection of plant species at the genus level. Owing to differences in social behavior, body size, and habitat selection, the three species further expanded their differentiation in resource selection, thereby utilizing environmental resources more efficiently. We believe that these are the main reasons for the stable coexistence of the three species in the study area. The Nyenchen Tanglha Mountain region in Tibet has abundant ungulate resources for two main reasons. First, the local area benefits from the advantageous steep mountainous terrain, rich forest resources, and suitable climatic conditions, which provide an ideal habitat for alpine musk deer, red serow, and white-lipped deer. Second, the local Tibetan people have a unique religious belief system and spontaneously engage in wildlife conservation, thereby reducing human disturbances and threats to wildlife populations. However, the local area has a large population of grazing livestock, consisting mainly of yaks and goats. Excessive grazing behavior may lead to increased inter-species competition, habitat destruction, and spread of infectious diseases among wild ungulates. To protect the local ungulate species, proper management and control measures must be implemented in the local grazing industry.

## Figures and Tables

**Figure 1 animals-14-02205-f001:**
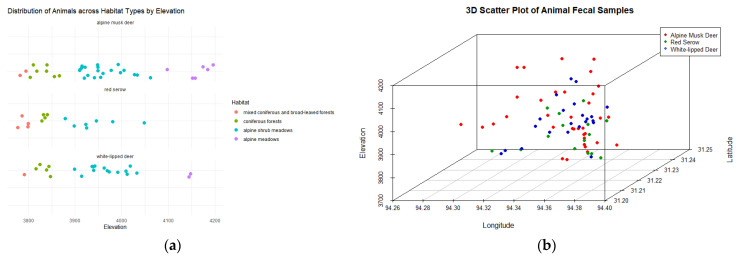
(**a**) Distribution of animals across habitat types by elevation. (**b**) 3D scatter plot of animal fecal samples.

**Figure 2 animals-14-02205-f002:**
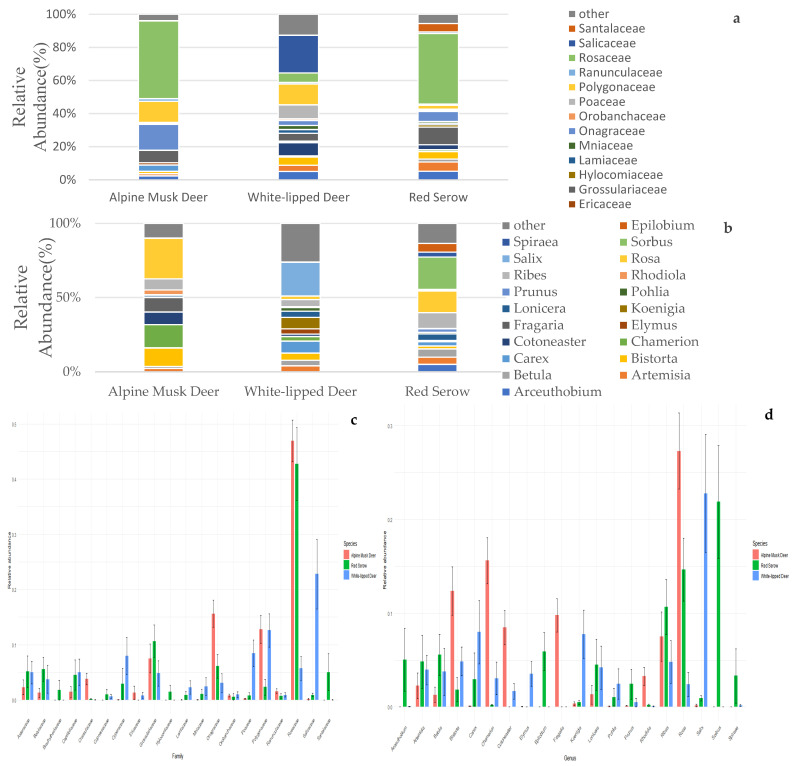
(**a**) Stacked bar plot showing the relative abundance of food at the family level for the three animal species. (**b**) Stacked bar plot showing the relative abundance of food at the genus level for the three animal species. (**c**) Dietary relative abundance at the family level. (**d**) Dietary relative abundance at the genus level.

**Figure 3 animals-14-02205-f003:**
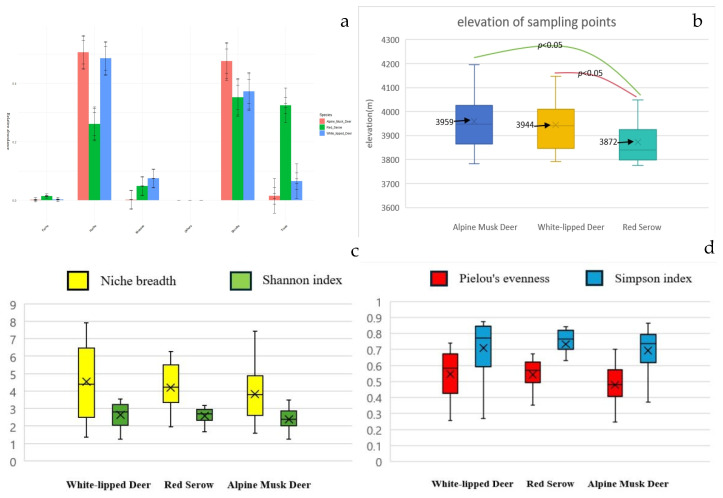
(**a**) Dietary types of alpine musk deer, white-lipped deer, and red serow. (**b**) Box plot of the elevation of sampling points. (**c**) Box plot of the ecological niche width and Shannon index of alpine musk deer, white-lipped deer, and red serow. (**d**) Box plot of Pielou’s evenness and Simpson indices of alpine musk deer, white-lipped deer, and red serow.

**Figure 4 animals-14-02205-f004:**
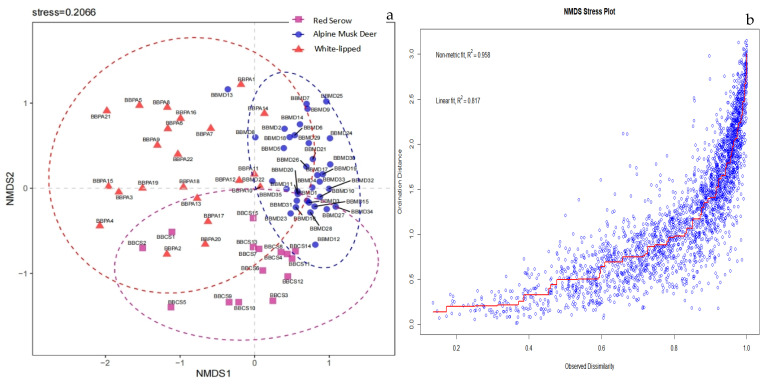
(**a**) Non-metric multidimensional scaling (NMDS) analysis of genus-level dietary composition among the three animal species. (**b**) Non-metric multidimensional scaling (NMDS) stress plot.

**Table 1 animals-14-02205-t001:** Comparison of ecological niche width and dietary diversity among alpine musk deer, white-lipped deer, and red serow (mean ± standard error).

Name	Alpine Musk Deer	White-Lipped Deer	Red Serow
Niche breadth	3.82 ± 0.25	4.53 ± 0.46	4.20 ± 0.33
Shannon index	2.38 ± 0.09	2.62 ± 0.15	2.57 ± 0.12
Pielou’s evenness	0.48 ± 0.02	0.55 ± 0.03	0.54 ± 0.03
Simpson index	0.69 ± 0.02	0.71 ± 0.04	0.73 ± 0.03

## Data Availability

The data presented in this study are openly available in NCBI at https://www.ncbi.nlm.nih.gov/ (accessed on 1 June 2024), BioProject number: PRJNA1125138.

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
