# Peer review of "Nutritional Partitioning among Sympatric Ungulates in Eastern Tibet"

_animals, 2024, doi:10.3390/ani14152205_

Round 1

Reviewer 1 Report

Comments and Suggestions for Authors

I appreciated the opportunity to review the manuscript entitled, “Foodplant DNA Barcoding Reveals Alpine Musk Deer Nutritional Partitioning with Sympatric Ungulates in Eastern Tibet.”  The stated objective of the paper was to determine if nutritional partitioning facilitates the coexistence of 3 ungulate species. I enjoyed reading the paper and overall, the writing was somewhat clear. I believe this is an important area of research and that there are novel data in this paper that would be of interest to the scientific community. However, I have a few general concerns with the manuscript in its current state that I believe must be addressed before I can make an informed recommendation on publication.

Frist, I think that the introduction is a little choppy and could do a better job leading the reader through the logic of introducing the objective (see comments below). Second, it is my opinion that an objective paragraph should have articulated, testable hypotheses. This paper provides an overall objective, but there are no hypotheses presented. Third, some areas of the methods provide detail in ad nauseam (e.g., description of the study site) whereas other areas are too brief to understand what was done and how (e.g., how the transects were identified, if transects were stratified by habitat, which analyses were used for which question/hypothesis, etc.). Further, it is my opinion that each paragraph of the methods and results should begin with a clear link to the questions being addressed in that paragraph. I have provided a few examples below. Finally, there are a few weaknesses in the presentation of the results that make it impossible to evaluate if the data support the conclusions. I believe the authors should provide the number of samples obtained from each species of ungulate along with an analysis of spatial overlap to illustrate if the animals are actually sympatric. In addition, estimates of error should accompany all estimates of relative abundance and diet composition so that the reader can assess the validity of the conclusions. Finally, rather than just reporting a P value with statements of results, list the analysis that was used and maybe even the degrees of freedom associated with the analysis so that the reader can determine the validity of the analysis.

In addition to these general concerns, I have provided several specific comments below that I believe will improve the manuscript.

I suggest changing the title to be more descriptive of the ecology. For example, “Nutritional partitioning among sympatric ungulates in eastern Tibet”. I do not like the current focus of the title on the method rather than the biology. Likewise, the study was focused on more than just alpine musk deer.

Line 32 – This first paragraph is comprised of several bold/factual statements that are each justified by a single citation—some of which are rather weak. I suggest toning down the boldness or bolstering the statements with corroborating references.

Line 64—The sentence beginning on this line seems disjunct and out of place. Needs better flow.

Line 69—The authors start a new topic here and therefore, should start a new paragraph. You could just delete the rest of this paragraph as you restate this information below.

Line 76—There are many factual statements in this paragraph without supporting literature. In my opinion, factual statements should be accompanied by supporting literature.

Line 89—In my opinion, the information from here forward should comprise a new paragraph.

Line 92—if the authors are going to state there are limitations, then they should provide the limitations.

Line 93—Instead of stating all the species that barcoding has been used for, just provide a few of the citations at the end of the previous sentence. The information in this sentence is not pertinent to this paper.

Line 100—Stating “in the same habitat” is redundant to stating they coexist. Delete the extra words that do not add.

Line100—You aimed at this objective or you investigated this objective? In my opinion, “aimed’ should be deleted.

Lines 101-105—In my opinion, there needs to be testable hypotheses in this paragraph that lead to an advancement of our understanding of ungulate coexistence.

Line 108—I believe a figure of the study area with a simple overlay of habitat type would be useful. Especially because I believe the authors need to provide evidence of sympatry or lack of spatial segregation to make their conclusion—nutritional partitioning facilitates coexistence.

Line 108-130—In my opinion, here and elsewhere in the manuscript are too wordy. Stick with the information to help the reader visualize the study site, but I do not think we need a detailed summary of what happens during each season unless necessary to understand the paper. Why list a few species and not all that also occur here? I can see listing species that potentially compete or predate, but listing otters seems completely unrelated?

Line 132—In my opinion, there should be a topic sentence that links the fact that fecal samples were collected to address one or more of the hypotheses listed in the introduction. As is, we don’t know why or from what species samples were collected.

Line 136—Was sampling stratified by habitat? How many transects in each habitat? Why were samples only collected over a single season and a single year? We samples cleared from the transects prior to the start so that timing of the scat was known?

Line 145—In my opinion, the topic sentence of each paragraph of the methods should be clearly linked to an hypothesis or a type of data required. Starting a paragraph stating “3 ml of fecal sample was transferred” does not lead the reader to what was done and why. In my opinion, this paragraph should begin with something like, “To determine the species of ungulate that produced each scat, we used….. This problem exists in most paragraphs of the methods.

Line 146—What is PBS? Acronyms should be spelled out at first usage. There are several instances in this manuscript where unaccompanied acronyms are used.

Line 169—What do you mean “fecal samples were classified”?

Line 224—Which analysis was used to address which hypothesis or question? Why multiple diversity indices?

Line 232—In my opinion, the results should begin with how many samples were collected from each species and each habitat. The reader should be convinced that all three deer species were sufficiently sampled and there is clear evidence for coexistence—not spatial partitioning that would also lead to differences in diet.

Line 232—Rather than start the result with a statement of the method, start the result paragraph with a biological result. Likewise, this was a 2.5 month study—in my opinion, stating a deer “consumes” rather than “during summer 2023 the deer consumed” is misleading. We do not know if diet is consistent through years or even among seasons. Stating that this is their diet is likely misleading.

Linese 248-287—All estimates of relative abundance and percent composition should be accompanied by an estimate of error. Assuming you had several samples from each deer and the deer were randomly distributed or at least similarly distributed in space, then estimates of error will help the reader interpret the significance of the results. Likewise, what exactly do the percentages mean? 46.9% of samples contained Rosaceae or 49.6% of extracted DNA was Rosaceae? If the latter, then we definitely need to have an estimate of error among individuals. For example, if one individual had 99% Rosaceae and two others had 25%, the mean would be 49.6%, but clearly, that number would have high variation with low confidence and may be misleading.

Line 278—This goes back to my last comment in the methods. Which analysis was used to determine significant differences for each question? Does this mean that the three species of deer consumed different species? Somewhere, I would like to see degrees of freedoms so that I can assess how the analyses were performed.

Lines 289-295—Again, I would like to see degrees of freedom from the analyses. Also and as stated above, all the percentages listed here should include an estimate of error among samples.

Line 320—This analysis was not presented in the methods section?

Discussion—I believe an analysis of spatial overlap is required to make the assertion that nutritional niche partitioning facilitates coexistence rather than simply spatial partitioning.

Author Response

For research article

Response to Reviewer X Comments

1. Summary 

Dear Reviewer,

Initially, I would like to express my sincere gratitude to you for the invaluable suggestions provided during the review of our manuscript. Your feedback has pointed out the areas that require further improvement, enabling us to comprehensively revise and refine the content of the paper.

In the Introduction section, we have removed erroneous statements, modified content that was overly rigid or disconnected, and added verifiable research hypotheses at the end of the paragraph to provide a clearer overall objective for the article, thereby improving the logic and fluency of the content.

In the Materials and Methods section, we have simplified the description of the study area, removing irrelevant information, revised inappropriate descriptions of the research methods, and added more detailed descriptions of the research methods to ensure that readers can better understand our research process.

In the Results section, we have removed content that could easily lead to controversy, revised the descriptions of sample size and analysis methods, and added analyses of the spatial distribution of the three ungulate species and the error analysis of the relative abundance data to strengthen the credibility of our conclusions.

In the Discussion section, we have reduced redundant and complex reiterations, and revised discussions that could be controversial.

Thank you for your invaluable review comments. Your professional advice will help me further refine the manuscript, making it more rigorous, clear, and convincing. I will thoughtfully incorporate your suggestions and expedite the revisions. If there is anything else that needs to be supplemented, I will be sure to communicate with you.

Wishing you all the best!

2. Questions for General Evaluation

Reviewer’s Evaluation

Response and Revisions

Does the introduction provide sufficient background and include all relevant references?

Can be improved

According to the reviewer's request, the research background has been revised.

Are all the cited references relevant to the research?

Yes/Can be improved/Must be improved/Not applicable

All references are relevant to this research.

Is the research design appropriate?

Can be improved

Based on the reviewer's request, improve the research design.

Are the methods adequately described?

Must be improved

To enhance the description of the research methods

Are the results clearly presented?

Must be improved

Explain the results of the study more clearly

Are the conclusions supported by the results?

Must be improved

The content of the results section has been added to better support the conclusions

3. Point-by-point response to Comments and Suggestions for Authors

Comments 1: I suggest changing the title to be more descriptive of the ecology. For example, “Nutritional partitioning among sympatric ungulates in eastern Tibet”. I do not like the current focus of the title on the method rather than the biology. Likewise, the study was focused on more than just alpine musk deer.

Response 1: Thank you for pointing this out. We agree with this comment. Therefore, we changed the title of the paper and the new title is “Nutritional partitioning among sympatric ungulates in eastern Tibet”. You can find this revision in the line 2 of page 1.

Comments 2: Line 32 – This first paragraph is comprised of several bold/factual statements that are each justified by a single citation—some of which are rather weak. I suggest toning down the boldness or bolstering the statements with corroborating references.

Response 2: Thank you for pointing this out. We agree with this comment. We have weakened the overly absolute expression in this paragraph. You can find this modification in line 34-48 of page 1-2.

Comments 3: Line 64—The sentence beginning on this line seems disjunct and out of place. Needs better flow.

Response 3: Thank you for pointing this out. We agree with this comment. We've revised this section to make it smoother. You can find this modification in line 63-67 of page 2.

Comments 4: Line 69—The authors start a new topic here and therefore, should start a new paragraph. You could just delete the rest of this paragraph as you restate this information below.

Response 4: Thank you for pointing this out. We agree with this comment.We restarted a new paragraph at lines 85 of page 2.

Comments 5: Line 76—There are many factual statements in this paragraph without supporting literature. In my opinion, factual statements should be accompanied by supporting literature.

Response 5: Thank you for pointing this out. We agree with this comment. In lines 85-96 of page 2, we have added new references

Comments 6: Line 89—In my opinion, the information from here forward should comprise a new paragraph.

Response 6: Thank you for pointing this out. We agree with this comment. We started a new paragraph on line 97 of page 3.

Comments 7: Line 92—if the authors are going to state there are limitations, then they should provide the limitations.

Response 7: Thank you for pointing this out. We agree with this comment. In lines 99-100 of page 3, we enumerate the limitations of traditional methods, including low resolution and high cost and effort.

Comments 8: Line 93—Instead of stating all the species that barcoding has been used for, just provide a few of the citations at the end of the previous sentence. The information in this sentence is not pertinent to this paper.

Response 8: Thank you for pointing this out. We agree with this comment. You can find this modification on lines 103 of page 3.

Comments 9: Line 100—Stating “in the same habitat” is redundant to stating they coexist. Delete the extra words that do not add.

Response 9: Thank you for pointing this out. We agree with this comment, in the same habitat already be deleted. You can find this modification on line 107 of page 3.

Comments 10: Line100—You aimed at this objective or you investigated this objective? In my opinion, “aimed’ should be deleted.

Response 10: Thank you for pointing this out. We agree with this comment, aimed already be deleted. You can find this modification on line 107 of page 3.

Comments 11: Lines 101-105—In my opinion, there needs to be testable hypotheses in this paragraph that lead to an advancement of our understanding of ungulate coexistence.

Response 11: Thank you for pointing this out. We agree with this comment. We hypothesize that the three species of ungulates exhibit distinct nutritional partitioning in their dietary. This nutritional partitioning serves as one of the mechanisms facilitating their coexistence  You can find this modification in lines 110-112 of page 3.

Comments 12: Line 108—I believe a figure of the study area with a simple overlay of habitat type would be useful. Especially because I believe the authors need to provide evidence of sympatry or lack of spatial segregation to make their conclusion—nutritional partitioning facilitates coexistence.

Response 12: Thank you for pointing this out. We agree with this comment. We have assessed the spatial overlap of the three species by quantifying the latitude, longitude, and elevation information of the sampling points within the study area. We have created a 3D scatter plot of the sampling points to visually represent this spatial overlap. Additionally, based on habitat photos and elevation data from the sampling points, we have preliminarily determined the habitat types and incorporated this information into the scatter plot. These measures provide preliminary evidence of sympatry among the three species, supporting our conclusion that nutritional partitioning facilitates coexistence. You can find these modifications in lines 258-282 of page 6.

Comments 13: Line 108-130—In my opinion, here and elsewhere in the manuscript are too wordy. Stick with the information to help the reader visualize the study site, but I do not think we need a detailed summary of what happens during each season unless necessary to understand the paper. Why list a few species and not all that also occur here? I can see listing species that potentially compete or predate, but listing otters seems completely unrelated?

Response 13: Thank you for pointing this out. We agree with this comment. We have reduced overly lengthy sections of the paragraph, retaining the environmental description during the study period. The list of species has been revised to exclude the otter, and now only includes species that may have competitive or predatory relationships with the study subjects within the research area. You can find these modifications in lines 115-130 of page 3.

Comments 14: Line 132—In my opinion, there should be a topic sentence that links the fact that fecal samples were collected to address one or more of the hypotheses listed in the introduction. As is, we don’t know why or from what species samples were collected.

Response 14: Thank you for pointing this out. We agree with this comment. To investigate whether the three ungulate species display significant dietary differences to alleviate competition pressures, we collected fecal samples from alpine musk deer, white-lipped deer, and red serow in the study area. These samples, gathered between July 20, 2023, and October 3, 2023, will help us test our hypotheses on dietary differentiation among these species. You can find these modifications in lines 132-138 of page 3.

Comments 15: Line 136—Was sampling stratified by habitat? How many transects in each habitat? Why were samples only collected over a single season and a single year? We samples cleared from the transects prior to the start so that timing of the scat was known?

Response 15: Thank you for pointing this out. Sample collection was not stratified by habitat but conducted along vertical transects from low to high elevations, covering major habitat types in the study area. Due to the high elevation of the research area, during cold climate periods, fecal samples can be easily covered by heavy snow, which would create significant difficulties and dangers for sample collection. Therefore, we have chosen to conduct the fecal sample collection during the periods of July 20, 2023, and October 3, 2023, in order to ensure the safety of the field personnel as well as the success rate of the sample collection. We collected fresh fecal samples, recorded their locations and timestamps to ensure the timeliness of dietary information and to avoid duplicate sampling. We plan to supplement future studies with fecal samples collected across multiple years and seasons to expand our sample size and temporal scope, thus enhancing our understanding of the dietary habits and ecological roles of these species comprehensively.

Comments 16: Line 145—In my opinion, the topic sentence of each paragraph of the methods should be clearly linked to an hypothesis or a type of data required. Starting a paragraph stating “3 ml of fecal sample was transferred” does not lead the reader to what was done and why. In my opinion, this paragraph should begin with something like, “To determine the species of ungulate that produced each scat, we used….. This problem exists in most paragraphs of the methods.

Response 16: Thank you for pointing this out. We agree with this comment. We have added the main paragraphs in the Methods section to clarify the objectives intended to be achieved or hypotheses to be tested through this method.  You can find these modifications in lines 132-138 of page 3, line 156-157, 182-184 of page 4, line 241-256 of page 5-6.

Comments 17: Line 146—What is PBS? Acronyms should be spelled out at first usage. There are several instances in this manuscript where unaccompanied acronyms are used. 

Response 17: Thank you for pointing this out. PBS stands for phosphate-buffered saline. We have revised several instances in the article where abbreviations were used without explanation. You can find these modifications in lines 158 of page 4.

Comments 18: Line 169—What do you mean “fecal samples were classified”?

Response 18: Thank you for pointing this out. "The fecal samples were classified" originally meant that the fecal samples, once species identification was successful, were organized for subsequent analysis. However, due to a misunderstanding caused by poor wording, I have deleted this potentially confusing content. You can find these modifications in lines 182-184 of page 4.

Comments 19: Line 224—Which analysis was used to address which hypothesis or question? Why multiple diversity indices?

Response 19: Thank you for pointing this out. We have added the specific issues or hypotheses addressed by each method in the Data Statistical Analysis section. Using different biodiversity indices allows us to assess the complexity and diversity of animal diets from multiple perspectives, such as species richness, dominance, and evenness. This helps to provide a more comprehensive understanding of the animals' dietary structure and preferences. You can find these modifications in lines 241-256 of page 5-6.

Comments 20: Line 232—In my opinion, the results should begin with how many samples were collected from each species and each habitat. The reader should be convinced that all three deer species were sufficiently sampled and there is clear evidence for coexistence—not spatial partitioning that would also lead to differences in diet.

Response 20: Thank you for pointing this out. We agree with this comment. As requested, we have started the results section by detailing the number of fecal samples collected from each species in each habitat. You can find these modifications in lines 259-266 of page 6.

Comments 21: Line 232—Rather than start the result with a statement of the method, start the result paragraph with a biological result. Likewise, this was a 2.5 month study—in my opinion, stating a deer “consumes” rather than “during summer 2023 the deer consumed” is misleading. We do not know if diet is consistent through years or even among seasons. Stating that this is their diet is likely misleading.

Response 21: Thank you for pointing this out. We agree with this comment. We pointed out in lines 284-285 of page 6 that "We plan to utilize DNA barcoding to reveal the dietary habits of three ungulate species within the study area over a 2.5-month period during the summer of 2023," to avoid any misunderstandings in the later content.

Comments 22: Linese 248-287—All estimates of relative abundance and percent composition should be accompanied by an estimate of error. Assuming you had several samples from each deer and the deer were randomly distributed or at least similarly distributed in space, then estimates of error will help the reader interpret the significance of the results. Likewise, what exactly do the percentages mean? 46.9% of samples contained Rosaceae or 49.6% of extracted DNA was Rosaceae? If the latter, then we definitely need to have an estimate of error among individuals. For example, if one individual had 99% Rosaceae and two others had 25%, the mean would be 49.6%, but clearly, that number would have high variation with low confidence and may be misleading.

Response 22: Thank you for pointing this out. We agree with this comment. Your suggestion is very meaningful. We have added the standard error of dietary among individuals in the Food Composition section. You can find these modifications in lines 294-299 of page 7.

Comments 23: Line 278—This goes back to my last comment in the methods. Which analysis was used to determine significant differences for each question? Does this mean that the three species of deer consumed different species? Somewhere, I would like to see degrees of freedoms so that I can assess how the analyses were performed.

Response 23: Thank you for pointing this out. We agree with this comment. The significant analysis methods we supplemented in the text. The degrees of freedom (df) information for these methods is:AMOVA and ANOVA:Between-group df = 2 (when comparing 3 species), Total df = 71. Kruskal-Wallis analysis: Between-group df = 2 (when comparing 3 species). ANOSIM (Analysis of Similarities) is a non-parametric, rank-based method, which typically does not provide degrees of freedom.

In general, when we are comparing the significant differences in data among three species, the between-group degrees of freedom is usually 2, and the total degrees of freedom is 71.

You can find these modifications in lines 241-256 of page 5-6, 335, 348, line 355-356 of page 8, line 389 of page 10, line 419, 422 of page 11.

Comments 24: Lines 289-295—Again, I would like to see degrees of freedom from the analyses. Also and as stated above, all the percentages listed here should include an estimate of error among samples.

Response 24: Thank you for pointing this out. We agree with this comment. Supplementary information on degrees of freedom and error estimates has been added; you can find modifications on lines 355-356 and 366 of page 8-9.

Comments 25: Line 320—This analysis was not presented in the methods section?

Response 25: Thank you for pointing this out. The individual specialization index is introduced in lines 248-250 of page 5-6.

Comments 26: Discussion—I believe an analysis of spatial overlap is required to make the assertion that nutritional niche partitioning facilitates coexistence rather than simply spatial partitioning.

Response 26: Thank you for pointing this out. We agree with this comment. In the results section, we analyzed the spatial overlap of the three animal species and provided preliminary evidence of sympatric distribution. You can find these contents in lines 258-282 of page 6.

4. Response to Comments on the Quality of English Language

Point 1: English language fine. No issues detected

Response 1: Thank you for your review. I'm pleased to hear that the English translation met the necessary standards. Your assessment is greatly appreciated.

5. Additional clarifications

[Here, mention any other clarifications you would like to provide to the journal editor/reviewer.]

Reviewer 2 Report

Comments and Suggestions for Authors

suggest some modification on this article.

(1) As DNA barcoding provide relative abundance, and the results are supposed to be influenced by digestibility of chloroplast in leaves especially during summer. So I suggest, even maybe limited, abundance data on direct field observation or browsing traces would be helpful as some kind of reference for food intake ranking.

(2) A map on geographic location of sampling together with GPS points of samples from the three target species are necessary, otherwise readers cannot tell how sympatric they are in a sampling period lasting 2.5 months.

(3) Some revisions on figures are needed. The barplot of diet composition for the three species is not quite clear, I recommend authors change the colors of each taxa unit and make the barplot alluvial one to make them more comparable and understandable.

(4) Dendrogram for sample cluster is not appropriate and validation for cluster results are needed. If NMDS have already been performed, sample cluster seem to express duplicated information.

(5) More information on NMDS besides stress value is needed, authors should provide figure of convergence of NMDS as well. It is ideal if more habitat information such as elevation could be fit on the NMDS figure.

Author Response

For research article

Response to Reviewer X Comments

1. Summary

Dear Reviewer,

Initially, I would like to express my sincere gratitude to you for the invaluable suggestions provided during the review of our manuscript. Your feedback has pointed out the areas that require further improvement, enabling us to comprehensively revise and refine the content of the paper.

In the Introduction section, we have removed erroneous statements, modified content that was overly rigid or disconnected, and added verifiable research hypotheses at the end of the paragraph to provide a clearer overall objective for the article, thereby improving the logic and fluency of the content.

In the Materials and Methods section, we have simplified the description of the study area, removing irrelevant information, revised inappropriate descriptions of the research methods, and added more detailed descriptions of the research methods to ensure that readers can better understand our research process.

In the Results section, we have removed content that could easily lead to controversy, revised the descriptions of sample size and analysis methods, and added analyses of the spatial distribution of the three ungulate species and the error analysis of the relative abundance data to strengthen the credibility of our conclusions.

In the Discussion section, we have reduced redundant and complex reiterations, and revised discussions that could be controversial.

Thank you for your invaluable review comments. Your professional advice will help me further refine the manuscript, making it more rigorous, clear, and convincing. I will thoughtfully incorporate your suggestions and expedite the revisions. If there is anything else that needs to be supplemented, I will be sure to communicate with you.

Wishing you all the best!

2. Questions for General Evaluation

Reviewer’s Evaluation

Response and Revisions

Does the introduction provide sufficient background and include all relevant references?

Yes

According to the reviewer's request, the research background has been revised.

Are all the cited references relevant to the research?

Yes/Can be improved/Must be improved/Not applicable

All references are relevant to this research.

Is the research design appropriate?

Yes

Based on the reviewer's request, improve the research design.

Are the methods adequately described?

Can be improved

To enhance the description of the research methods

Are the results clearly presented?

Can be improved

Explain the results of the study more clearly

Are the conclusions supported by the results?

Yes

The content of the results section has been added to better support the conclusions

3. Point-by-point response to Comments and Suggestions for Authors

Comments 1: (1) As DNA barcoding provide relative abundance, and the results are supposed to be influenced by digestibility of chloroplast in leaves especially during summer. So I suggest, even maybe limited, abundance data on direct field observation or browsing traces would be helpful as some kind of reference for food intake ranking.

Response 1: I greatly appreciate you raising this valuable suggestion. Your concern about the potential influence of chloroplast digestibility on the DNA barcoding analysis results is completely valid, and this is indeed an important factor that I had overlooked in the process of writing the paper.

I fully agree with your perspective of incorporating direct field observations or trace surveys as a complement to the DNA barcoding analysis. While DNA barcoding can provide relative abundance information, as you mentioned, these data may be affected by seasonal factors, especially during the summer. Therefore, the integration of multiple survey methods will help improve the reliability and comprehensiveness of the research results.

I will seriously consider your suggestion in my future research work. In the next stage of the study, I will attempt to cross-validate the field observation data with the DNA barcoding analysis results, in order to better understand the food intake behavior. This will help refine my research methodology and achieve more accurate and comprehensive research outcomes.

I thank you again for raising this valuable suggestion. Your advice is of great value in improving the quality and innovative capability of my research. I will treat it seriously and strive to put it into practice in my future work. If there are any other areas that need improvement, please continue to provide your valuable guidance.

Comments 2: (2) A map on geographic location of sampling together with GPS points of samples from the three target species are necessary, otherwise readers cannot tell how sympatric they are in a sampling period lasting 2.5 months.

Response 2: Thank you for your valuable feedback. I appreciate your suggestion regarding the need for a map showing the geographic location of the sampling sites, along with the GPS coordinates of the samples collected for the three target species. This information is crucial to allow readers to understand the degree of sympatry among the species during the 2.5-month sampling period.

In response to your comment, I would like to provide the following expanded reply:

I have included the geographic location information and GPS coordinates of the sampling sites in the results section of the paper. Specifically, I have provided a 3D scatter plot that shows the latitude, longitude, and elevation of the sampling points. This visual aid allows readers to clearly see the spatial distribution of the samples collected for the three target species.

Furthermore, I have also included a habitat type plot that overlays the sampling point locations.

By providing these spatial data visualizations, I aim to give readers a comprehensive understanding of the degree of sympatry among the three target species within the sampling area and timeframe. This information is essential for interpreting the results and assessing the validity of the conclusions drawn from the DNA barcoding analysis.

Please let me know if you have any other suggestions for improving the presentation of the geographic context and sample locations in the paper. I am happy to further refine the response to ensure the reviewer's concerns are adequately addressed.

You can find these revisions in lines 258-282 of page 6.

Comments 3: (3) Some revisions on figures are needed. The barplot of diet composition for the three species is not quite clear, I recommend authors change the colors of each taxa unit and make the barplot alluvial one to make them more comparable and understandable.

Response 3:Thank you very much for your suggestion regarding the revisions needed for the figures. I fully understand your concern about the lack of clarity in the bar plot of diet composition for the three species. I have supplemented the original stacked bar plot with additional bar plots (Figures 2.c, 2.d) that show the feeding on different plant genera for each species along the x-axis. These two new figures can more clearly display the selection of each animal for different plants, making up for the lack of clarity in the original diet composition bar plot.

If you have any other suggestions for improving the presentation of the figures, I would be more than happy to receive your further feedback. I will continue to strive to enhance the readability and information conveyance of the figures, ensuring the best visual support for both the reviewers and the readers.

You can find these revisions in lines 292-299 of page 7.

Comments 4: (4) Dendrogram for sample cluster is not appropriate and validation for cluster results are needed. If NMDS have already been performed, sample cluster seem to express duplicated information.

Response 4: Thank you very much for your suggestions regarding the modifications to the figures. I understand your concerns about the use of the dendrogram for sample clustering and the need for validation of the clustering results. Your feedback is extremely valuable.

Based on your recommendation, I have decided to remove the dendrogram, as the NMDS analysis has already reflected the similarities between the samples. Retaining the dendrogram may result in duplicated information, which would be detrimental to the conciseness and focus of the article.

I will carefully consider your other improvement suggestions and do my best to enhance the presentation quality of the figures, ensuring that I provide clear and valuable visual information for both the reviewers and the readers. Once again, thank you for your review and corrections, as they are extremely helpful in improving the quality of my work.

Comments 5: (5) More information on NMDS besides stress value is needed, authors should provide figure of convergence of NMDS as well. It is ideal if more habitat information such as elevation could be fit on the NMDS figure.

Response 5: Thank you for your valuable suggestions. In this study, we provided the NMDS confidence ellipse plot (Figure 4a). We further calculated the non-metric fit (r^2=0.958) and linear fit (r^2=0.817) of the NMDS analysis. These very high fit values indicate that our NMDS analysis has very good quality, as it has effectively captured the non-linear relationships between the samples and to some extent preserved the linear relationships as well.

Your suggestion to supplement more habitat information provides us with a good research direction. We will collect richer habitat-related data in future studies to further improve our understanding of the community structure and its relationship with the environment.

Thank you again for your valuable feedback. We will carefully incorporate your suggestions and strive to improve our research work. Please feel free to let us know if you have any other questions.

You can find these revisions in lines 382-391 of page 9-10.

4. Response to Comments on the Quality of English Language

Point 1: I am not qualified to assess the quality of English in this paper

Response 1: we have carefully reviewed the manuscript ourselves and made necessary adjustments to improve readability and coherence.

5. Additional clarifications

Reviewer 3 Report

Comments and Suggestions for Authors

Dear  Authors,

Please find uploaded the pdf file of the reviewed manuscript with my many comments.

The study is interesting and based on a modern methodology, but you should improve the description of the procedures and at some parts the structure of the manuscript.

Best wishes

Comments on the Quality of English Language

Author Response

For research article

Response to Reviewer X Comments

1. Summary

Dear Reviewer,

Initially, I would like to express my sincere gratitude to you for the invaluable suggestions provided during the review of our manuscript. Your feedback has pointed out the areas that require further improvement, enabling us to comprehensively revise and refine the content of the paper.

In the Introduction section, we have removed erroneous statements, modified content that was overly rigid or disconnected, and added verifiable research hypotheses at the end of the paragraph to provide a clearer overall objective for the article, thereby improving the logic and fluency of the content.

In the Materials and Methods section, we have simplified the description of the study area, removing irrelevant information, revised inappropriate descriptions of the research methods, and added more detailed descriptions of the research methods to ensure that readers can better understand our research process.

In the Results section, we have removed content that could easily lead to controversy, revised the descriptions of sample size and analysis methods, and added analyses of the spatial distribution of the three ungulate species and the error analysis of the relative abundance data to strengthen the credibility of our conclusions.

In the Discussion section, we have reduced redundant and complex reiterations, and revised discussions that could be controversial.

Thank you for your invaluable review comments. Your professional advice will help me further refine the manuscript, making it more rigorous, clear, and convincing. I will thoughtfully incorporate your suggestions and expedite the revisions. If there is anything else that needs to be supplemented, I will be sure to communicate with you.

Wishing you all the best!

2. Questions for General Evaluation

Reviewer’s Evaluation

Response and Revisions

Does the introduction provide sufficient background and include all relevant references?

Yes

According to the reviewer's request, the research background has been revised.

Are all the cited references relevant to the research?

Yes/Can be improved/Must be improved/Not applicable

All references are relevant to this research.

Is the research design appropriate?

Yes

Based on the reviewer's request, improve the research design.

Are the methods adequately described?

Must be improved

To enhance the description of the research methods

Are the results clearly presented?

Can be improved

Explain the results of the study more clearly

Are the conclusions supported by the results?

Yes

The content of the results section has been added to better support the conclusions

3. Point-by-point response to Comments and Suggestions for Authors

Comments 1: You could add species names not involved in the title, i.e.

red serow, and white-lipped deer 

Response 1: Thank you for pointing this out. We agree with this comment. We have added Alpine Musk Deer, White-lipped Deer, and Red Serow to the keywords. You can find this modification in lines 31-32 of page 1.

Comments 2: maybe "variety" would be better instead of "abundance", since here it relates to the number of species

Response 2: Thank you for pointing this out. We have made modifications to the content. You can find this change in lines 43 of page 1.

Comments 3: I would add the latin names here as this is the first mentioning of the species name in the main text

Response 3: Thank you for pointing this out. We agree with this comment. We have added the Latin names of the three ungulate species. You can find this modification in lines 77-79 of page 2.

Comments 4: You should move this part, line 68-75 to the last paragraph of the introduction, since here it is still not understandable why you are mentioning those three species. Start with the general description than in the next step turn to your study region and specific study species.

Response 4: Thank you for pointing this out. We agree with this comment. We have made revisions to this content, which you can find in lines 71-73 of page 2.

Comments 5: better could be "observers" or "examiners" 

Response 5: Thank you for pointing this out. We agree with this comment. You can find this modification at line 93 of page 2.

Comments 6: add Tibet

Response 6: Thank you for pointing this out. We agree with this comment. You can find this modification at line 106 of page 3.

Comments 7: better would be "could be"

Since dietary overlap itself is not a proof of competition, you should also prove resource limitation, which could be done experimentally. 

Response 7: Thank you for pointing this out. We agree with this comment. You can find this modification at line 108 of page 3.

Comments 8: You should rewrite this part of sentence.

Dietary overlap itself is not a proof of competition, you should also prove resource limitation, which could be done experimentally (you decrease the density of one species or you add some food etc.)

Response 8: Thank you for pointing this out. We agree with this commen. We have rewritten this sentence. You can find it at lines 105-112 of page 3.

Comments 9: I would delete this last sentence, this will be described in the methods. Here it would raise questions without answer, e.g. why only between July and October and not seasonally.

Response 9: Thank you for pointing this out. We agree with this comment. We have deleted this section. t was originally located at line 112 of page 3.

Comments 10: You could add map, but also photo

Response 10: Thank you for pointing this out. We agree with this comment. We have assessed the spatial overlap of the three species by quantifying the latitude, longitude, and elevation information of the sampling points within the study area. We have created a 3D scatter plot of the sampling points to visually represent this spatial overlap. Additionally, based on habitat photos and elevation data from the sampling points, we have preliminarily determined the habitat types and incorporated this information into the scatter plot. These measures provide preliminary evidence of sympatry among the three species, supporting our conclusion that nutritional partitioning facilitates coexistence. You can find these modifications in lines 258-282 of page 6.

Comments 11: willow species (Salix sp.)

Response 11: Thank you for pointing this out. We agree with this comment. You can find these modifications in lines 130 of page 3.

Comments 12: You should explain why you did not perform a seasonally repeated sample collection to cover the whole year, why this small temporal window can be enough or what were your limitations to be able to carry out the research beyond this interval.

(I guess it is impossible to collect sample in other periods there?)

Response 12: Thank you for pointing this out. We agree with this comment . Due to the high elevation of the research area, during cold climate periods, fecal samples can be easily covered by heavy snow, which would create significant difficulties and dangers for sample collection. Therefore, we have chosen to conduct the fecal sample collection during the periods of July 20, 2023, and October 3, 2023, in order to ensure the safety of the field personnel as well as the success rate of the sample collection.sure the safety of team members and minimize unnecessary environmental impact. You can find these modifications in lines 134-138 of page 3.

Comments 13: Were not there clear differences in the shape or size of the droppings of these three ungulates helping field identification?

Response 13: Thank you for pointing this out. We agree with this comment. Based on our collected experience and experimental results, Alpine musk deer exhibit fixed defecation points in the wild. Their fecal samples are spindle-shaped with smaller pellets compared to white-lipped deer and red serow. They are visually distinguishable from these two ungulate species. However, the feces of white-lipped deer and red serow are similar in appearance, resembling dates when fresh, making it difficult to visually differentiate between the two. DNA extraction is required for species identification in these cases.

Comments 14: Then you should state here that you were collecting reference plant materials, since here you start with the mentioning of the equipment but nothing about the aims.

Response 14: Thank you for pointing this out. We agree with this comment. In order to provide some reference materials for the subsequent identification of dietary plants, we have also collected plant samples from the vicinity of the fecal sampling sites. You can find these modifications in lines 149-151 of page 4.

Comments 15: What is the meaning of this? All faecal samples of the same ungulate species were merged and used as a single composite sample?

Therefore you obtained three composite samples in total, one for each ungulate species?

Please, clarify this detail.

Response 15: Thank you for pointing this out. "The fecal samples were classified" originally meant that the fecal samples, once species identification was successful, were organized for subsequent analysis. However, due to a misunderstanding caused by poor wording, I have deleted this potentially confusing content.

Comments 16: At the beginning you should state what was the final sample size for the three different ungulate species, respectively, based on your DNA-based species identification procedure for the consumer species.

Response 16: Thank you for pointing this out. We agree with this comment. You can find these modifications in lines 259-266 of page 6.

Comments 17: It is not entirely clear how it was calculated.

About which are you speaking?

You should state about it and explain it in the Methods!

Relative abundance = abundance of a species a in a sampling unit/total abundance of all species in a sampling unit.

Relative frequency = number of sampling units a species occurred/total number of sampling units

Response 17: Thank you for pointing this out. We agree with this comment. We supplemented the explanation of the calculation method for relative abundance in lines 245-247 of page 5.

Comments 18: Sorry, but without any explanation and description what to see on the figure it is not self-explanatory for Readers. You should add the description of results to Fig 1c in the text.

E.g. finally there were 72 faecal samples posssible to identify to ungulate species? It has not been mentioned until now.

Response 18: Thank you for pointing this out. We agree with this comment. Considering the overlap in effect between Figure 2c and the NMDS plot, which could lead to confusion, we have decided to remove Figure 2c. You can find these modifications in lines 294-295 of page 7.

Comments 19: I do not understand, how it is possible relative to the above-mentioned sentence? (line 234-235)

Alpine musk deer consumes a diet consisting of plants belonging to 74 families and 144 genera

Would it mean that the 94,6/92% encompasses only 9 and 12 of 74 and 144 families and genera, respectively?

Please, clarify!

Response 19: Thank you for pointing this out.

 First, it is important to note that the study of the alpine musk deer's diet was based on sample surveys and analyses, rather than a detailed statistical record of its entire dietary range. This means that the data we obtained is based on local samples and may not fully capture the complete dietary preferences of the alpine musk deer across its entire habitat.

In this context, we observed that the alpine musk deer's diet is predominantly concentrated on plants belonging to 9 families and 12 genera, which account for 94.6% and 92% of its total dietary intake, respectively. This result is reasonable and reflects the following points:

The alpine musk deer may have a greater preference for certain plant families and genera, as these provide its primary sources of nutrition. This selective foraging behavior is a common ecological adaptation strategy observed in many animals.

Although the alpine musk deer's total dietary range encompasses 74 families and 144 genera, not all plant species hold an equally significant place in its diet. These 9 families and 12 genera are likely the alpine musk deer's most common and preferred food sources.

The local samples may not fully cover the entirety of the alpine musk deer's dietary range, but they have captured its primary nutritional sources. The data on relative abundance can reflect the main dietary characteristics of the alpine musk deer in the studied region.

Therefore, although the alpine musk deer's total dietary range includes 74 families and 144 genera, the highest relative abundances are observed in the 9 families and 12 genera, which account for a significant portion of its diet. This is a reasonable research finding that can provide valuable insights into the alpine musk deer's dietary preferences and ecological adaptations.

Comments 20: Start a new paragraph here for a better structure of the text

Response 20 Thank you for pointing this out. We agree with this comment. You can find these modifications in lines 329 of page 8.

Comments 21: ungulates or herbivores

Response 21: Thank you for pointing this out. We agree with this comment. You can find these modifications in lines 330-331 of page 8.

Comments 22: between what? ungulates or food plants? Clarify this statement

Response 22: Thank you for pointing this out. We agree with this comment. You can find these modifications in lines 333-335 of page 8.

Comments 23: I would recommend summarizing those data in a Table or Figure

Response 23: Thank you for pointing this out. We agree with this comment. The data for this section can be referenced in Figure 2d on line 294 of page 7.

Comments 24: rephrase this

It is not the elevation of faecal samples but the elevation of sample point locations

Response 24: Thank you for pointing this out. We agree with this comment. You can find these modifications in lines 354 of page 8.

Comments 25: Add axis title

Response 25: Thank you for pointing this out. We agree with this comment. You can find these modifications in lines 366 of page 9.

Comments 26: Trees not trees (if upper case is used for other categories)   

Response 26: Thank you for pointing this out. We agree with this comment. You can find these modifications in lines 366 of page 9.

Comments 27: Is this correct like this?

Better could be "have more opportunities to choose different foods"   

Response 27: Thank you for pointing this out. We agree with this comment. You can find these modifications in lines 407-408 of page 10.

Comments 28: the environment is not coexisting, the species are coexisting in the same environment

Response 28: Thank you for pointing this out. We agree with this comment. You can find these modifications in lines 413 of page 10.

Comments 29: I do not think that it is a good solution to repeat the results in a so long paragraph again (line 355-377). This is the Discussion section.

Response 29: Thank you for pointing this out. We agree with this commen.. We have already reduced the content. You can find these modifications in lines 416-432 of page 11.

Comments 30: I think you should start here, and delete or shorten a lot the text in the previous 20 lines.  

Response 30: Thank you for pointing this out. We agree with this comment. We have already reduced the content. You can find these modifications in lines 416-432 of page 11.

Comments 31: Be careful with this statement or improve it!

When considering energy requirements relative to body size, smaller animals actually have higher metabolic rates. This means that per unit of body weight, smaller animals require more energy than larger animals.

Response 31: Thank you for pointing this out. We agree with this comment. We removed the discussion on energy and nutritional requirements and instead analyzed the feeding quantity of animals of different body sizes. The revised sentence is as follows: White-lipped deer and red serow have significantly larger body sizes than alpine musk deer. Larger body size means they need to consume a greater quantity of food. The amount provided by just a few dietary items would be insufficient to meet the feeding requirements of the larger-bodied animals. Therefore, white-lipped deer and red serow tend to select a more diverse array of food sources as their primary dietary components, in order to fulfill their nutritional needs. You can find these modifications in lines 460-466 of page 11.

4. Response to Comments on the Quality of English Language

Point 1: Minor editing of English language required

Response 1:  Thank you very much for the valuable feedback from the reviewer. I will carefully consider and edit the English language usage to meet the journal's language requirements.

I will thoroughly review the sentence structure, word choice, grammar, and spelling to ensure the English expression is more fluent and accurate.

I will focus on addressing any specific issues or areas for improvement pointed out by the reviewer and optimize those sections accordingly.

While preserving the core content, I will try to rephrase the ideas using more natural and academic language to better align with the format expectations for an SCI journal paper.

If necessary, I will reorganize the paragraph structure to ensure the overall presentation is logically clear and well-structured.

Finally, I will meticulously proofread the entire text to confirm the language is smooth, the phrasing is appropriate, and it meets the editor's expectations.

5. Additional clarifications
